



# On how sediment supply affects step formation, evolution and stability in steep streams: an experimental study

Matteo Saletti[1] and Marwan A. Hassan[1]

[1]Department of Geography, The University of British Columbia, 1984 West Mall, Vancouver, BC, V6T1Z2, Canada

**Correspondence:** Matteo Saletti (matteo.saletti@ubc.ca)

**Abstract.** We present results from an experimental campaign run in a steep flume subject to longitudinal width variations and different sediment feed rates. The experiments were designed to study how sediment supply influences step formation, step location, and step stability. Our results show that steps are more likely to form in narrowing areas (i.e., where the channel width is getting smaller moving downstream) because of particle jamming, and these steps are also more stable. Sediment supply increases particle activity generating a more dynamic channel morphology with more steps forming and collapsing. However, sediment supply does not inhibit step formation, since more steps are generated in experiments with sediment feed than without it. Time-series of step formation, evolution, and destruction show that the maximum number of steps is achieved for average values of sediment supply. We summarize this outcome in a conceptual model where the dependence of step frequency on sediment supply is expressed by a bell curve. Sediment yield measured at the channel outlet followed the sediment feed at the inlet closely, even when we fed 50% more and 50% less than the transport capacity. This outcome challenges the applicability of the concept of transport capacity to steep channels and highlights the key role played by sediment supply for channel stability and sediment transport. Finally, we detected a positive correlation between sediment concentration and step destruction, which highlights the key role played by granular jamming for step formation and stability.

## 1 Introduction

Step-pool channels are often found in steep mountain streams, when large boulders and woody debris jam in the transverse direction, forming steps followed by pools of finer sediment (Chartrand et al., 2011; Grant et al., 1990; Montgomery & Buffington, 1997). This morphology has been extensively studied because it is visually appealing, and provides a good habitat for fish and an effective tool for energy dissipation that keeps the channel stable even at high flows (see reviews by Chin and Wohl, 2005; Church and Zimmermann, 2007; Comiti and Mao, 2012). For these reasons, step-pool channels are often artificially designed in stream restoration projects (e.g., Comiti et al., 2009b; Yu et al., 2010), instead of more impactful infrastructures made of concrete, such as check dams (Chin et al., 2009; Piton et al., 2017). Therefore, geomorphologists and engineers require understanding of the conditions under which steps form, remain stable, and destabilize.

Several studies in the last decades have increased our knowledge on how step-pool systems function, especially with regard to the step-forming mechanisms (Chin, 1999; Curran, 2007; Golly et al., 2019; Saletti et al., 2016; Zimmermann et al., 2010), the stability of steps (Waters and Curran, 2012; Zhang et al., 2019, 2018; Zimmermann et al., 2010), the links between channel



and hillslope dynamics (Golly et al., 2017; Molnar et al., 2010), and the relations between flow magnitude, flow resistance and sediment transport (Comiti et al., 2009a; Hohermuth & Weitbrecht, 2018; Saletti et al., 2015; Turowski et al., 2009; Zimmermann, 2010). Field studies and flume experiments highlighted how boulder protrusion (Yager et al., 2018, 2007), grain clustering (Johnson, 2017), and the supply of fine sediment (Johnson et al., 2015) impact flow resistance and therefore channel

stability and sediment transport in step-pool channels. Furthermore, the importance of granular interactions for step formation and stability has been previously recognized (Church and Zimmermann, 2007; Saletti et al., 2016) and it has been suggested that steps are more stable than predicted for the emergence of force-chains in the transversal direction that keep them in place even when subjected to higher shear stress (Bouchard et al., 2001; Church and Zimmermann, 2007; Saletti and Hassan, 2020). The basic questions of how and where steps form and under which conditions remain stable is still paramount for practitioners

who are often asked to design steps (or similar structures) to stabilize steep channels while maintaining their ecological value and visual appeal (e.g., Chin et al., 2009; Thomas et al., 2000). Existing design criteria consider only flow variables (e.g., flow rate and flow depth), grain-size and channel geometry, ignoring factors that might strongly impact the stability of artificial step-pools (e.g., sediment supply and longitudinal width variations).

More recently, Golly et al. (2019) and Saletti and Hassan (2020) showed how longitudinal variations in channel width

regulate both the process leading to step formation and the locations where steps preferentially form. More specifically, the experiments of Saletti and Hassan (2020) demonstrated that steps formed by particle jamming in narrow and especially narrowing locations tend to be more frequent and more stable. A limitation of these experiments is that they were conducted in absence of sediment feed, a condition that is not always realistic in mountain streams, especially in those coupled with active hillslopes (e.g., Recking et al., 2012; Turowski et al., 2009) or linked to a source of sediment such as a melting glacier (e.g., Comiti at

al., 2019). Sediment supply has been shown to be a very important control on the development and evolution of bedforms in gravel-bed streams (e.g., Hassan et al., 2020; Hassan and Church, 2000; Venditti et al., 2017), but no direct study has addressed the impact of different sediment feed rates on step formation and evolution in a steep channel subject to longitudinal width variations.

To address this issue, we ran experiments with the same flume geometry and sediment mixture used by Saletti and Hassan

(2020) but we feed sediment at different rates and for different flow discharges, in order to study how steps develop and evolve under different sediment supply regimes. At the end of each experimental run, we turned off the sediment supply and increased the flow rate until the bed was completely scoured, to assess channel stability in sediment-starved conditions. Field evidence (Recking et al., 2012) shows that step-pool channels directly connected to sediment supply sources are less stable, and previous research (e.g., Chin, 1998; Curran, 2007; Saletti et al., 2016; Zimmermann et al., 2010) suggested that low sediment supply

is necessary for step stability, because a high sediment supply would bury the steps. More recently, Waters and Curran (2012) run experiments with different sediment and water discharges finding a complex relationship between sediment supply, flow resistance and step stability. In one of the most cited conceptual models for step stability, the jammed-state hypothesis proposed by Church and Zimmermann (2007), one of the three parameters that control step stability is the sediment concentration, defined by the authors as the ratio between sediment supply and water discharge. They hypothesized that the sediment concentration would need to be small in order to achieve step stability, since large sediment supply would bury the steps. Our experiments,





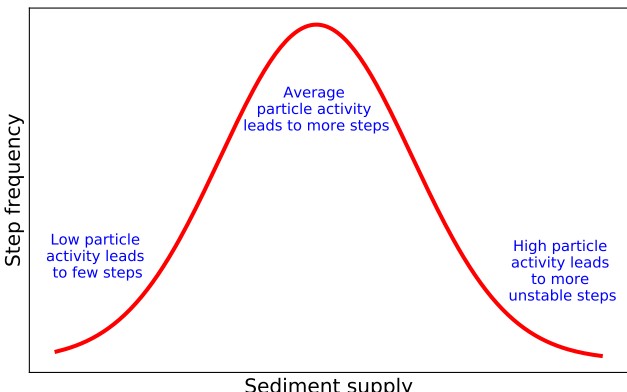

**Figure 1.** Conceptual model that shows step frequency as a function of sediment supply (i.e., particle activity).

run keeping all conditions the same except for the feed rate, can be used to test this specific hypothesis. Our expectation is that large sediment supply would also increase particle activity (i.e., the degree of interaction between the different particles in motion and between these particles and the bed) which could possibly enhance both step formation due to granular interactions, and step instability due to grain dislodgment.

We frame our study in terms of three main research hypotheses. (1) With sediment supply narrowing areas should generate more steps because of particle jamming enhanced by a larger particle activity (Golly et al., 2019; Saletti and Hassan, 2020). (2) An increasing sediment feed rate should cause more sediment transport and a more dynamic channel, therefore leading to increasing chances of step collapse. (3) The relation between step frequency and sediment feed should be described qualitatively by a curve qualitatively similar to that shown in Figure 1, where step frequency should be small when sediment supply is too

low (for the absence of enough particles to form steps) or too high (for too much particle activity leading to step collapse and burial), while reaching a maximum for intermediate values of sediment supply.

    The specific research questions we aim to answer are: (1) Does sediment feed rate influence the frequency and location of steps in steep streams? (2) Does the stability of steps (i.e., their survival time) depend on the sediment supply regime and/or the step location? (3) How do the outcomes of no-feed experiments compare to those of feed experiments in terms of step

frequency, location and dynamics? Given the increasing changes in sediment supply regimes due to urbanization and climate change and the widespread use of step-pool channels in stream restoration projects, we believe this is a timely and important topic for both river scientists and practitioners.

## 2   Experimental Setup and Methods

We performed three experiments (Exp 50, Exp 100, and Exp 150) in the Mountain Channel Hydraulic Experimental Laboratory

at The University of British Columbia, using the same flume geometry and sediment grain size distribution as in Saletti and Hassan (2020). We used a 5-m long, 0.5-m wide and 1-m deep flume at a slope of 8%. We included trapezoidal concrete



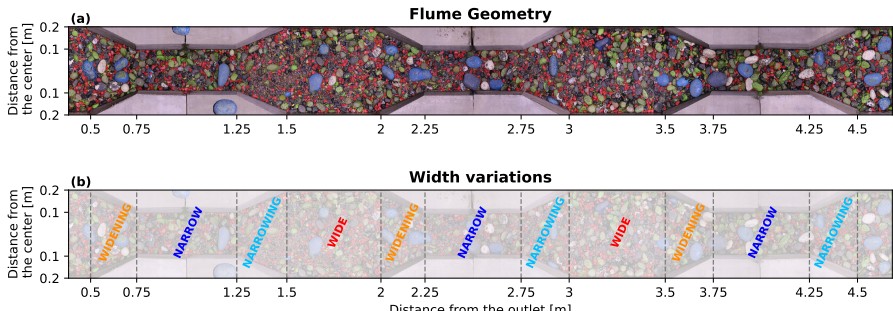

**Figure 2.** (a) Top view of the flume geometry used in the experiments. Grains of different sizes were painted in different colors to facilitate grain-size recognition and surface structure detection. (b) Channel locations of longitudinal width variations. The flow is right to left.

elements (as shown in Fig. 2) to create narrow and wide areas (of 20 cm and 40 cm, respectively), in addition to narrowing and widening areas.

After one hour of conditioning flow (q=5 l/s/m) which produced little or no sediment transport, the three experiments were
subjected to the same step-wise increasing flow rates (increased by 20% each hour as done by Saletti and Hassan, 2020) but different sediment feed rates. We estimated the sediment feed rates using the Wilcock and Crowe (2003) sediment transport model for the bulk grain size distribution and the different flow rates. These values constituted the feed rates for experiment 100 (as to 100% feed capacity); the feed rates for experiment 50 and 150 were obtained by multiplying the feed rates of experiment 100 by 0.5 and 1.5 respectively. The three experiments were meant to represent three different sediment supply regimes, rather
than simulate precise values of transport capacity. During each hour, the amount of sediment was fed over the first 40 minutes with a conveyor belt located at the flume inlet; the grain-size distribution of the sediment feed was the same as the bulk one used for the bed. After 7 hours, we turned off the supply of sediment and increased the flow rate by 20% each hour until the bed was scoured, completely exposing the bottom of the flume in at least one location. We refer to the feed phase in the 3 experiments as 50F, 100F and 150F, while we refer to the subsequent no feed phase as 50N, 100N and 150N. Table 1 and
Figure 3 report the flow and sediment feed rates used in the three experiments. The starting value of flow rate is larger than that used in Saletti and Hassan (2020) because the three lowest flow rates used there did not produce significant changes.

Before starting each experiment, the flume was filled with a 15-cm deep layer of sediment whose bulk grain-size distribution was the same used by Saletti and Hassan (2020), having $d_{50}$=15mm, $d_{16}$=3mm, and $d_{84}$=29mm. The mixture included sediment sized between 0.5 mm and 64 mm, which was divided into 14 $\psi/2$ classes. Stones of different classes were painted in different
colors to facilitate surface grain-size analyses.

Every hour the flow was stopped to collect topographic and grain-size data using a green laser and a camera mounted on the top a moving chart. Digital elevation models of the bed surface were obtained from the laser at 2-mm resolution in the horizontal direction and 1-mm resolution in the vertical direction. Photos were used to estimate bed grain-size distribution and identify keystone location as explained in Saletti and Hassan (2020). A uniformly spaced grid of 200 points was overlapped





**Table 1.** Values of unit discharge $q$ and sediment feed rate $Q_s$ used in the experiments

| Exp_hour | $q$ $[l/s/m]$ | $Q_s$ $[kg/h]$ | Exp_hour | $q$ $[l/s/m]$ | $Q_s$ $[kg/h]$ | Exp_hour | $q$ $[l/s/m]$ | $Q_s$ $[kg/h]$ |
|---|---|---|---|---|---|---|---|---|
| 50F_01 | 17.5 | 2.9 | 100F_01 | 17.5 | 5.8 | 150F_01 | 17.5 | 8.7 |
| 50F_02 | 21.0 | 6.6 | 100F_02 | 21.0 | 13.2 | 150F_02 | 21.0 | 19.8 |
| 50F_03 | 25.0 | 13.7 | 100F_03 | 25.0 | 27.3 | 150F_03 | 25.0 | 41.0 |
| 50F_04 | 30.0 | 26.8 | 100F_04 | 30.0 | 53.6 | 150F_04 | 30.0 | 80.4 |
| 50F_05 | 36.0 | 47.8 | 100F_05 | 36.0 | 95.5 | 150F_05 | 36.0 | 143.3 |
| 50F_06 | 43.2 | 83.8 | 100F_06 | 43.2 | 167.5 | 150F_06 | 43.2 | 251.3 |
| 50F_07 | 51.9 | 138.8 | 100F_07 | 51.9 | 277.6 | 150F_07 | 51.9 | 416.4 |
| 50N_08 | 51.9 | 0 | 100N_08 | 51.9 | 0 | 150N_08 | 51.9 | 0 |
| 50N_09 | 51.9 | 0 | 100N_09 | 62.3 | 0 | 150N_09 | 62.3 | 0 |
| 50N_10 | 62.3 | 0 | 100N_10 | 62.3 | 0 | 150N_10 | 62.3 | 0 |
|  |  |  | 100N_11 | 74.7 | 0 |  |  |  |

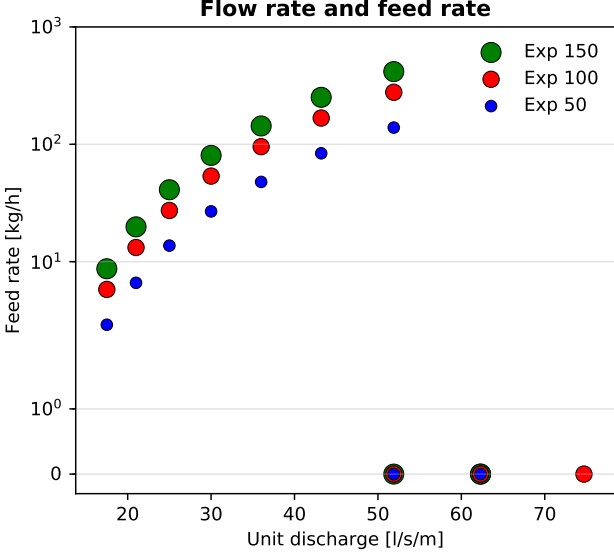

**Figure 3.** Flow rates and feed rates used in the 3 experiments. The marker size is proportional to the feed rate. In the last hours of each experiment (see Table 1) we increased the flow rate without feeding sediment.

to the picture and grains at each node were manually identified. Sediment transport rates were measured at the flume outlet at 1-Hz resolution for the 14 grain size fractions with a light table (Zimmermann at al., 2008a).



To reduce the level of subjectivity often associated to bedform identification, steps were extracted from digital elevation models using a scale-free rule-based algorithm (Saletti and Hassan, 2020; Zimmermann at al., 2008b) that accounts for spatial variability of step structures. Only steps that occupied more than half of the total channel width were mapped.

## 3    Results

The experiments conducted with sediment supply produced more steps than those without sediment supply (Saletti and Hassan, 2020) but higher feed rates decreased the average number of steps. Narrowing areas formed steps preferentially, where large particles were often deposited and jammed. Step formation/collapse and local scour/deposition continually occurred. Sediment feed enhanced particle activities for all grain sizes, increasing the frequency of particle-particle and particle-bed interactions, apparently raising the propensity for jamming. Even experiment 150, where we fed 50% more of the evaluated transport capacity, did not show significant sediment aggradation. Light table measurements of sediment transport rates demonstrated that sediment yield matched sediment feed quite closely. Despite large particle activity, localized areas of scour/deposition, and bed surface structure break-up/formation, sediment transport demonstrated steady-state characteristics, with sediment feed and yield differing by at most 10%. Sediment concentration $c_s$ clearly played a role in terms of steps stability, as more steps collapsed with increasing values of $c_s$.

In the next sections, we present results from experiments 50, 100 and 150 (hereafter referred to as "feed experiments") and from the experiment reported in Saletti and Hassan (2020) (hereafter referred to as "no-feed experiments") in terms of (a) step frequency, (b) step location, (c) step dynamics, (d) grain-size and sediment yield, and (e) sediment concentration and step stability.

## 3.1    Step frequency

The steps were identified and tracked from digital elevation models and images in the experiments with the algorithm described in Saletti and Hassan (2020). All the feed experiments generated consistently a step-pool morphology with the number of steps depending on both flow and feed rates. Experiment 50 had an average numbers of 9.4 steps per hour (10.3 during the feed part and 7.3 during the no feed part), experiment 100 had an average number of 8 steps per hour (9.1 during the feed part and 6 during the no feed part), experiment 150 had an average number of 6.4 steps per hour (7.6 during the feed part and 3.7 during the no feed part). As a comparison, the no-feed experiments (i.e., experiments $N_{3a}$ and $N_{3b}$ in Saletti and Hassan, 2020) had an average number of 5.4 steps. The relationship between the number of steps and flow rate for the feed experiments is shown in Fig. 4, together with the no-feed experiments.

Two main points can be made. (1) There is a large variability in the number of steps: step numbers range between 3 and 14 in the feed experiments, and between 3 and 8 in the no-feed experiments; this suggests a very dynamic channel morphology, a point that will be explored in the next sections. (2) The number of steps in the feed experiments clearly decreases with both flow rate (i.e., moving left to right in the x axis in Fig. 4) and feed rate (i.e., moving top to bottom in the y axis in the 3 different



Earth **Surface**
**Dynamics**
Discussions

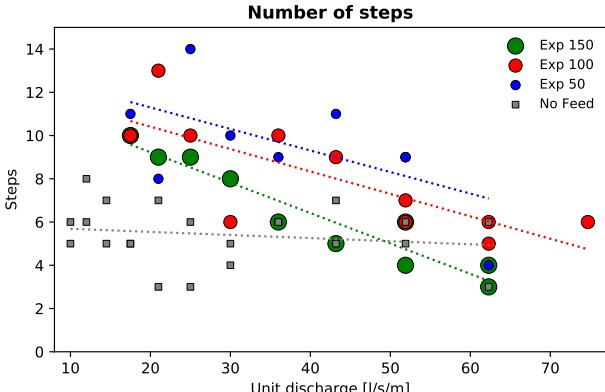

**Figure 4.** Number of steps detected in the 3 experiments at the end of each 1-hour run, shown with circles, and in the 2 experiments without sediment feed from Saletti and Hassan (2020), shown with squares. The size of the circles is proportional to the sediment feed rate. Dashed lines are the best-fit lines for the 3 feed experiments; the values of $R^2$ are 0.36, 0.60, and 0.91 for experiments 50, 100, and 150 respectively. The best-fit line for the no-feed experiments is almost flat but with $R^2 < 0.1$.

time series in Fig. 4); this decreasing trend becomes clearer as the feed rate increases, as suggested by the $R^2$ values. On the other hand, in the no-feed experiments there is no clear trend between the number of steps and the flow rate.

## 3.2 Step location

The no-feed experiments of Saletti and Hassan (2020) clearly showed that steps are more likely to form in narrow and narrowing locations rather than in wide and widening locations. The feed experiments reported here show a very similar result (Fig. 5): steps in narrow and narrowing areas (Fig. 5a, b) are generally more common than steps in wide and widening areas (Fig. 5c, d). The number of steps formed by particle jamming in narrowing areas (Fig. 5b) remains high and larger than zero throughout all the experiments, while steps in other locations display a decreasing trend with time and flow rate (Fig. 5a, c, d). As it was observed in the no-feed experiments, the number of steps in narrow areas (Fig. 5a) is quite high at the beginning for low values of flow rate but then it decreases as narrow areas (whose unit discharge is 2 times larger than that in wide areas) are subject to more erosion.

The likelihood of steps forming in certain locations can be explored by comparing the fraction of steps in a certain location with the fraction of channel length occupied by those locations, as done in Saletti and Hassan (2020). Fractions of steps differing significantly from fractions of areas indicate that width variations matter and that certain locations are more likely than others to have steps. More specifically, step fractions significantly larger than area fractions indicate a high likelihood to find steps in that location, while step fractions significantly smaller than area fractions indicate a low likelihood to find steps in that location.

Width variations control step location, with narrowing areas being the predominant place where steps form and remain stable (Fig. 6), as detected in the no-feed experiments (Saletti and Hassan, 2020). The comparison between step fractions and area

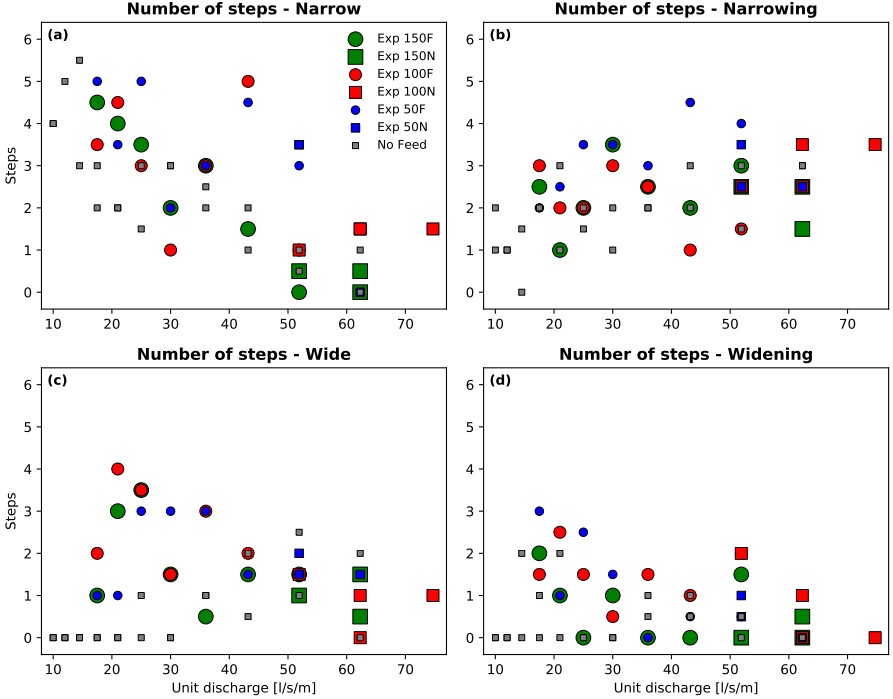

**Figure 5.** Number of steps detected in the feed and in the no-feed experiments at the end of each hour in (a) narrow areas, (b) narrowing areas, (c) wide areas, and (d) widening areas. Portions of experiments with sediment feed are shown with circles, while portions without sediment feed are shown with squares.

fractions yields three main results. (1) Narrowing areas are much more likely than others to have steps, and this effect becomes more pronounced as flow rates increases (as it can be inferred by the trend in Fig. 6b). (2) Steps in narrow areas are slightly more common for low to moderate values of flow rate, but this effect vanishes as flow rates increases and sediment feed is

160 turned off (as it can be inferred by the trend in Fig. 6a). (3) Steps are much less likely to form in wide and widening areas (e.g., most markers are plotting below the dashed line in Fig. 6c, d).

These trends in the number of steps and their location can be connected to the step-forming mechanisms due to a competition of granular and fluid forces (Saletti and Hassan, 2020). Narrow and narrowing areas are characterized by a larger bed shear stress but also a lower jamming ratio, and steps there form because of particle jamming. In wide and widening areas, the shear

165 stress decreases but the jamming ratio increases, and steps there form due to particle deposition around keystones (see also Golly et al., 2019). The results of the feed experiments confirm that jamming steps are more frequent than depositional steps, reinforcing that granular interactions are a key process to explain the occurrence and location of steps in steep channels, as previously suggested (e.g., Church and Zimmermann, 2007; Saletti et al., 2016; Saletti and Hassan, 2020; Zimmermann et al., 2010).



Earth **Surface**
**Dynamics**
Discussions

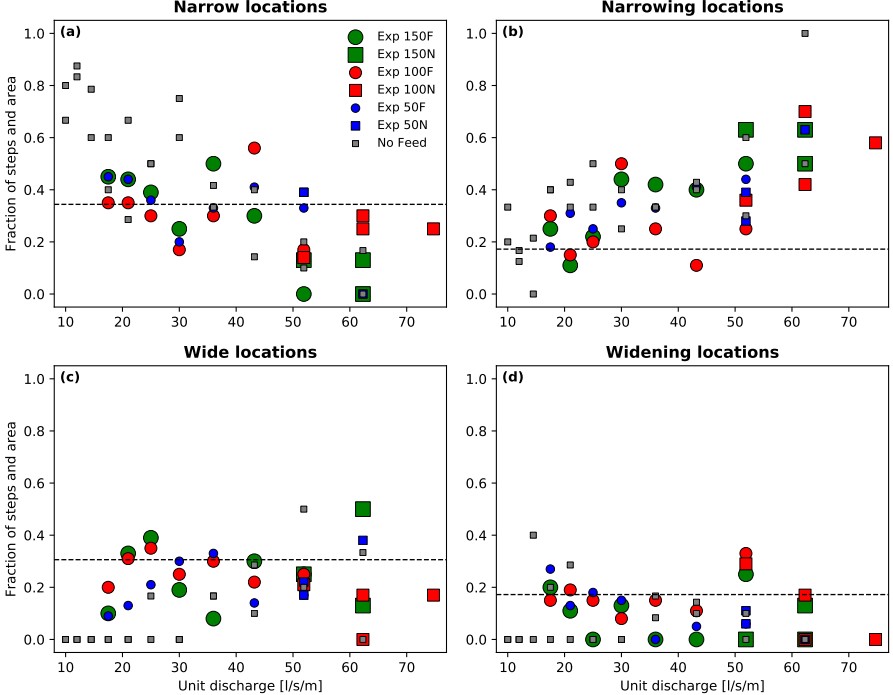

**Figure 6.** Fraction of steps detected in the feed and no-feed experiments at the end of each hour in (a) narrow areas, (b) narrowing areas, (c) wide areas, and (d) widening areas. Portions of experiments with sediment feed are shown with circles, while portions without sediment feed are shown with squares. The fraction of the flume occupied by those areas is shown with a dashed black line for comparison. Markers plotting above the dashed line indicate steps are more likely to occur in these location, markers plotting below that line indicate instead that steps are less likely to occur in these locations.

## 3.3 Step dynamics

The step identification algorithm coupled with the digital elevation models and the images allowed us to track individual steps and their evolution throughout the experiments. To better describe step dynamics, for each time interval we categorize each step into one of five groups: (1) steps newly formed, (2) steps that expanded in the transversal direction, (3) steps that contracted in the transversal direction, (4) steps that remained the same, and (5) steps that were destroyed. We show the temporal trends of these categories in Fig. 7.

A few points can be made. (1) All experiments were very dynamic, as very few steps remained the same (Fig. 7e). (2) Step formation and destruction were predominant during the feed phase and became less important when sediment feed was cut off (Fig. 7a, b); (3) Step expansion and contraction varied considerably in all the runs, both with flow and sediment feed rate (Fig. 7c, d).

To assess the stability of channel morphology in a different way, we grouped together steps that were newly formed and destroyed, as are proxy for channel instability, and steps that expanded, contracted and stayed the same, as a proxy for channel

Earth **Surface**
**Dynamics**
Discussions

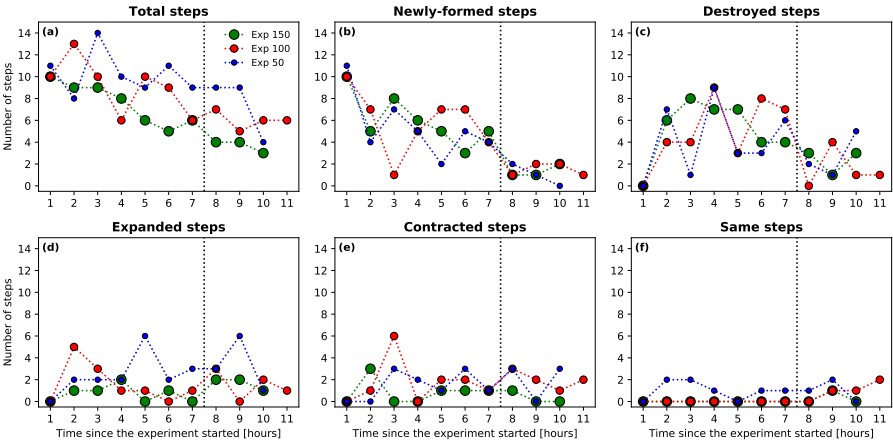

**Figure 7.** Step activity in the feed experiments expressed as number of (a) total steps, (b) newly-formed steps, (c) destroyed steps, (d) steps that have expanded, (e) steps that have contracted, and (f) steps that remained the same. The vertical dashed line separates the feed periods (on the left) from the no-feed periods (on the right).

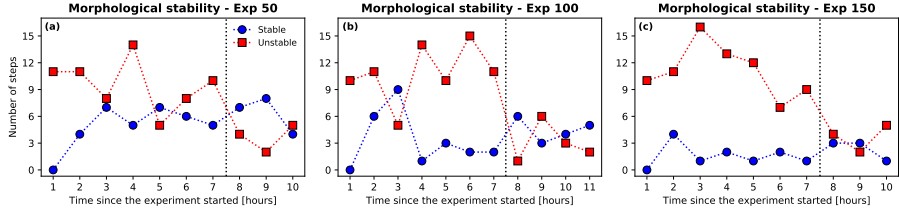

**Figure 8.** Stability of the step-pool morphology in time expressed as comparison between number of steps that formed an destroyed (i.e., unstable in red squares) and number of steps that remained the same or expanded/contracted (i.e., stable in blue circles) in (a) experiment 50, (b) experiment 100, and (c) experiment 150. The vertical dashed line separates the feed periods (on the left) from the no-feed periods (on the right).

stability. We show the temporal trends of these categories in Fig. 8. Two observations emerge from this analysis of morphological stability. (1) During the feed part of the experiments instability (i.e., step formation and destruction) was predominant, but, as soon as the sediment supply was shut off, the modification of pre-existing steps became more important (i.e., the two time-series in Fig. 8 became closer). (2) The difference between the instability and the stability of steps depends on sediment supply, as the two lines during the feed phase become more separated as the feed rate increases. This separation indicates that larger sediment supply enhances step formation and destruction. Cutting off the sediment supply suppresses step formation and destruction, making changes to existing steps the predominant process of morphological change.

The stability of steps can be explored also in terms of step survival time (i.e., for how many consecutive hours a step remains stable), and its dependence on the key variables explored in this study: the location within the channel and the magnitude of sediment supply. In Figure 9 we show violin plots of survival times as a function of (a) step location and (b) feed rate. In the





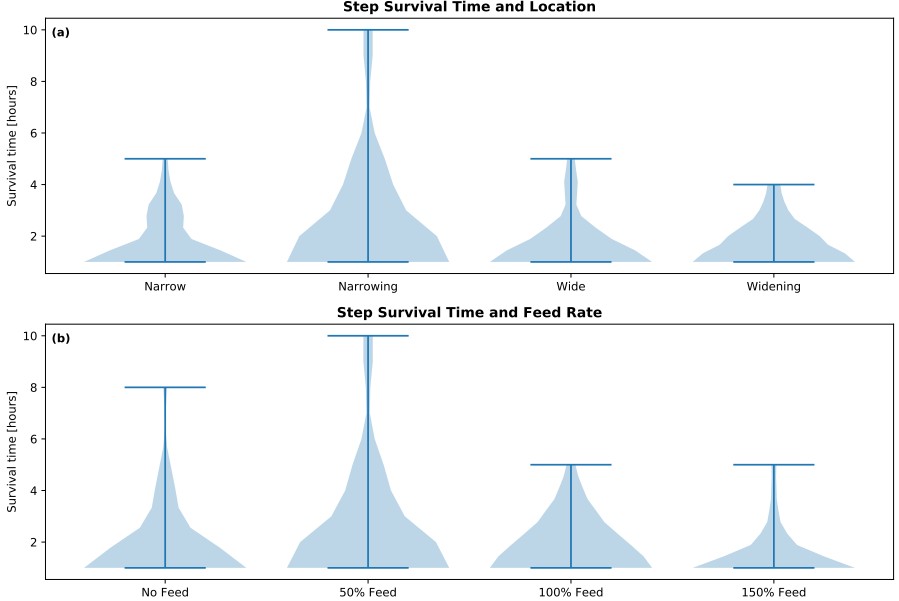

**Figure 9.** Violin plots of step survival times as a function of (a) step location and (b) sediment feed rate, with no-feed experiments shown for comparison.

case of step locations, the distribution of survival times demonstrates that steps in narrowing locations are more stable than those in other locations (Fig. 9a). None of the steps in narrow, wide, and widening locations survived for more than 5 hours, whereas a few steps in narrowing locations survived for the total duration of the experiment (i.e., 10 hours). With respect to the relationship between survival times and feed rate (Fig. 9b), the maximum survival time occurs for experiment 50, which has low sediment supply but larger than zero. Both experiments with no-feed and with larger feed rate have smaller survival times. These trends suggest that step stability is enhanced by average values of sediment supply, an interesting point that will be addressed in the discussion.

### 3.4 Grain size and sediment yield

The grain-size distribution of the bed was obtained by analyzing the images and manually identifying 200 stones on a uniformly spaced grid. The time series of $d_{16}$, $d_{50}$, and $d_{84}$ of the bed surface for the feed experiments are shown in Figure 10. The $d_{50}$ and (for the most part) the $d_{16}$ did not change much during either of the 3 experiments, whereas the $d_{84}$ was more dynamic, showing a consistent coarsening which became sharper around 4-6 hours into the experiments.

The sediment supply versus transport highlights that sediment yield measured at the flume outlet is dictated by the sediment feed at the inlet, as it can be seen in Figure 11. This confirms that sediment supply is a crucial control on sediment transport and yield. However, the equilibrium between sediment input and output does not mean that the channel morphology did not change, as it can be seen by looking at step dynamics in Fig. 7-8. It is worth nothing that we evaluated the feed rates to be used



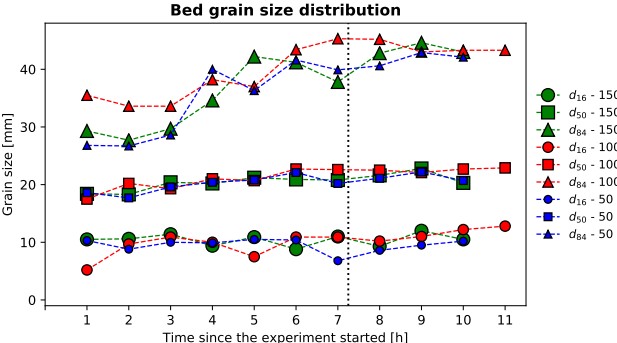

**Figure 10.** Grain-size distribution of the bed surface for the feed experiments. The $d_{16}$ is shown with circle, the $d_{50}$ with squares, the $d_{84}$ with triangles. The size of the markers is proportional to the feed rate. The vertical dashed line indicates when the sediment supply was cut off.

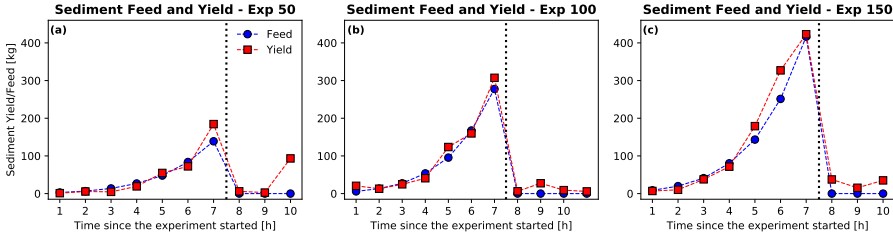

**Figure 11.** Sediment yield (red squares) measured at the channel outlet and sediment feed (blue circles) at the channel inlet in experiment (a) 50, (b) 100, and (c) 150. The vertical dashed line indicates when the sediment supply was cut off.

in the experiments by calculating the channel transport capacity with the Wilcock and Crowe (2003) model. Given that we ran only one experiment (exp 100) with these actual values, we expected net degradation in experiment 50 (where we fed 50% less of the calculated transport capacity) and net aggradation in experiment 150 (where we fed 50% more of the calculated transport capacity). This did not occur, since the sediment transport rate at the flume outlet tracks the feed rate at the flume inlet closely. Surprisingly, in experiment 150 there is more degradation than aggradation, as it can be seen in Fig. 11c, where sediment yields is almost always equal or larger than the sediment feed. The implications of this outcome will be discussed in the next sections.

## 3.5 Sediment concentration and step collapse

Considering that (a) we know the amount of sediment supplied to the channel, (b) we measured the amount of sediment that left the channel, and (c) we tracked all the steps that have been destroyed, it is possible to directly test the hypothesis proposed by Church and Zimmermann (2007) that a small sediment concentration is necessary for step stability. We calculated the sediment concentration $c_s$ in terms of both sediment feed ($c_{s,feed} = Q_{s,input}/Q$) and sediment yield ($c_{s,yield} = Q_{s,output}/Q$) as the



Earth **Surface**
**Dynamics**
Discussions

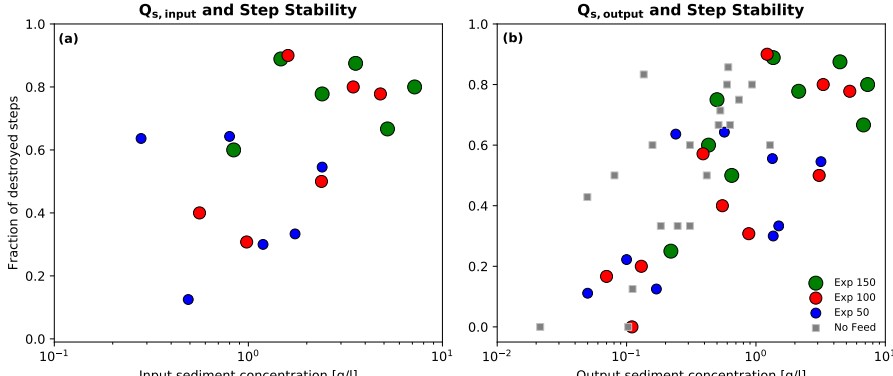

**Figure 12.** Fraction of steps that were destroyed in each run as a function of the logarithm of sediment concentration of (a) sediment input and (b) sediment yield. Feed experiments are shown with circles (whose size is proportional to the feed rate), and no-feed experiments are shown with squares.

ratio between solid and liquid discharge, and compared it with the fraction of steps that were destroyed in each run (number of steps destroyed at time $t$ divided by the total number of steps at time $t-1$). We show these trends in Fig. 12, where sediment concentrations are plotted in log scale.

The fraction of steps being destroyed clearly increases with increasing sediment concentration, as assumed in the jammed-state hypothesis by Church and Zimmermann (2007). In the feed experiments, the trend is clearer when plotted using the sediment concentration of the output ($R^2 = 0.54$ in Fig. 12b) than the sediment concentration of the input ($R^2 = 0.28$ in Fig. 12a). Unlike what proposed in the jammed-state hypothesis, we suggest that $c_{s,yield}$ should be used instaed of $c_{s,feed}$, at it is a better proxy for the particle activity that lead to step instability.

## 4    Discussion

### 4.1    Width variations and step location

In the no-feed experiments of Saletti and Hassan (2020), we demonstrated that longitudinal variations in channel width form steps in narrowing and narrow locations because of particle jamming, and that high flow rates scour away steps in narrow locations while preserving steps in narrowing locations. This outcome brought us to conclude that granular forces are predominant over fluid forces, since more steps occur in narrowing/narrow locations where the shear stress is larger and jamming ratio is smaller. The same behaviour can be observed in the feed experiments reported here: steps in narrowing locations are even more likely to occur than in the experiments without sediment supply (Fig. 6), and they are definitely more stable than steps in all other locations (Fig. 9). We attribute this outcome to the step-forming mechanism. Steps in narrowing locations are created by particle jamming enhanced by geometrical constraints, similarly to granular materials in a hopper (To et al., 2001): this process is strongly dependent on particle activity (i.e.,,, the more particle that are in transport, the more likely they are to jam) which is

a direct consequence of sediment supply. Therefore, we should expect that when sediment supply is large enough to maintain
an active level of sediment transport, more particle will jam, and more steps will be found in areas of narrowing. This outcome
is consistent with the reduced-complexity modelling results of Saletti et al. (2016), where the frequency of jamming events
creating steps has been shown to be directly dependent on particle activity and sediment supply. More generally, this highlights
the importance of considering granular effects and granular interactions to properly describe sediment transport and channel
morphology in steep coarse-bedded streams (Booth et al., 2014; Ferdowsi et al., 2017; Frey & Church, 2011).

These repeated experimental observations have important implications for step-pool design and mountain channel stability.
They suggest that to increase the success of stream restoration projects aimed at enhancing channel stability, steps should be
built where natural width constrictions favour keystone jamming. These jamming steps are not only more frequent, but also
more stable as shown in Fig. 6 and 9.

### 4.2   Step dynamics and channel stability

By tracking individual steps and their evolution, we demonstrated that step-pool channels are more dynamic when subjected to
sediment supply. During the feed experiments, when sediment supply was fed the dominant processes were step formation and
collapse, whereas when the sediment supply was cut off, contraction and expansion of existing steps became important (Fig.
7-8). These results are in agreement with the field study of Recking et al. (2012), who showed that natural step-pool channels
directly connected to sediment sources have less stable steps (they are destabilized by smaller floods).

Our experiments showed that step instability (i.e., more steps collapsing) triggered by large sediment supply does not nec-
essarily mean fewer steps, as the increased particle activity implies more step formation. For example, Figure 4 shows how
in some instances, experiments 150 had more steps than experiments 50 and 100, despite the larger sediment supply. This is
an important point, as it highlights how channel stability and morphological stability can be quite different in steep mountain
streams.

We found that increasing sediment concentration increased the chance of step collapse (Fig. 12), in agreement with the
jammed-state hypothesis (Church and Zimmermann, 2007). Based on our results, we propose that sediment concentration in
this context should be measured with respect to the sediment output rather than the input, since values of sediment yield are
a better proxy for the degree of a stability of the channel. Experiments conducted by Waters and Curran (2012) did not find a
consistent relationship between sediment concentration and step collapse, although their study considered temporal stability of
sequences rather than individual steps, and their flow rates remained constant for a longer period of time.

### 4.3   What maximizes step frequency?

Our experiments showed that the number of steps generated in feed experiments is larger than otherwise equivalent no-feed
experiments (Saletti and Hassan, 2020). However, during feed experiments the average number of steps detected at the end of
each hour decreased with feed rate (both in the same experiment and between different experiments). These results, combined
with the step dynamics shown in Fig. 8-9 and the increased particle activity with increasing sediment supply, suggest that
there is an "optimum" level of sediment supply that maximizes the number of steps. We hypothesize that the relation between



Earth **Surface**
**Dynamics**
Discussions

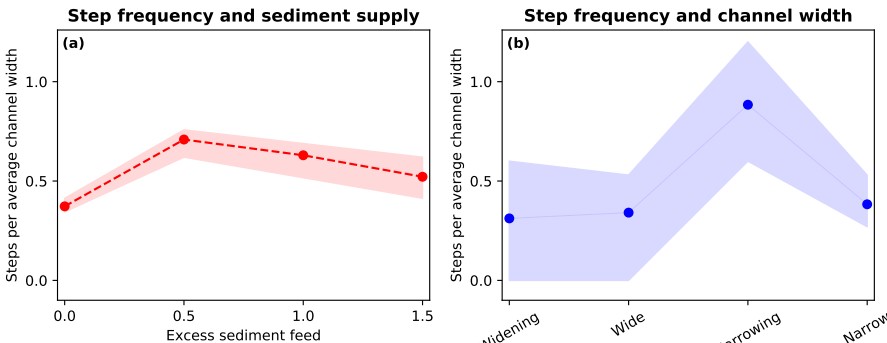

**Figure 13.** Step frequency (expressed as number of steps per number of channel widths) plotted as a function of (a) sediment supply and (b) longitudinal channel width variation. Dashed lines represent mean values, and the shaded area is the 25-75 percentile. The excess sediment feed in (a) is expressed as the sediment supply normalized by the transport capacity.

step frequency and sediment supply can be conceptually expressed by a curve similar to that displayed in Fig. 1, where the optimum (i.e., the maximum number of steps) is achieved for values of sediment supply larger than zero but smaller than the evaluated transport capacity. For values of sediment supply below this optimum level there is not enough sediment available

to build this maximum number of steps (i.e., particle activity is too low). Instead, for values of sediment supply above the optimum the system becomes too active (i.e., particle activity is too high), so that many steps forms but they are immediately destroyed, as Fig. 8 demonstrates. Increasing values of sediment supply (measured in terms of sediment concentration at the flume outlet) clearly showed an increasing fraction of steps destroyed (Fig. 12). However, the larger particle activity also increases the chances of step formation because of granular interactions, yielding the relationship displayed in Figure 1, where

the maximum number of steps is achieved for average values of sediment feed.

Using the data from both the no-feed and the feed experiments, we show how step frequency changes as a function of sediment supply and location in terms of channel width (Figure 13). To generalize our results, we show step frequency as number of steps per reach length expressed as number of average channel widths. We quantify the sediment supply with excess sediment feed, defined as the ratio between the sediment feed and the transport capacity (evaluated in our case with the model

of Wilcock and Crowe, 2003). In Figure 13 we plot both the average values from all our experiments and the 25 and 75 percentiles to show the variability around these values. As discussed before, step frequency has a maximum for average values of sediment supply (i.e., when the sediment feed is half of the transport capacity) and in narrowing locations. The comparison between Figure 13a and b suggests that the effect of width variations on step frequency is much stronger than that of sediment supply, although also much more variable, as the values of the percentiles suggest.

It is important to note that sediment feed rate and longitudinal channel width variations are only two of the variables that influences step frequency. Other variables are expected to be important, such as flow rates and the hydrological regime (e.g., Zhang et al., 2019, 2018), grain-size distribution for the availability of large grains acting as keystones (e.g., Hohermuth & Weitbrecht, 2018), channel geometry and slope (Chartrand et al., 2011). Sediment supply can also vary not only with respect to





the magnitude, but also in terms of duration and frequency (e.g., Hassan et al., 2020) . Especially in steep mountain channels,
where the supply of sediment is often episodic, these aspects should be further investigated. Finally, channel width variations in
steep channels are expected to occur in a less systematic way than those designed in our experiments, as well as with different
angles and potentially with different material.

### 4.4 What is transport capacity in steep channels?

The values of sediment yield measured at the channel outlet during the experiments were very similar to the values of sediment
feed imposed at the channel inlet (Fig. 11). The feed rates chosen in the three experiments spanned one order of magnitude and
were below, equal, and above the calculated capacity, yet the sediment yield was still determined by the supply. This suggests
that: (1) the channel adjusts its morphology to be able to carry the imposed load, and (2) the concept of transport capacity is
hardly applicable in steep mountain channels where the sediment transport rates are a function of the imposed feed and changes
in channel morphology (e.g., Saletti et al., 2015) more than anything else.

It is important to note that this might be a consequence of the (small) length of the flume: changes in channel morphology that
can store and release the imposed sediment supply could require much longer flumes to be captured in physical experiments.
Our flume length was ~15 times the average channel width, a measure that is usually considered to be enough to represent
a channel reach (Montgomery & Buffington, 1997, suggest for example 10-20 channel widths); however these results would
need to be tested in longer flumes in order to determine whether the pattern observed here depends on flume length or not.

### 4.5 Outlook

Our results highlight the important control of sediment supply and channel width variations on step formation, evolution,
and stability of steps in steep streams. However, some aspects deserve for sure further investigations. First, these experiments
were conducted under constant feed rate, a condition that is not always realistic in mountain streams, where mass movements
and climate variability often make the supply of sediment highly variable. Therefore, experiments should be conducted with
episodic sediment supply, to check whether this will change the outcome. Second, other variables are likely to be important,
such as channel slope, angle of channel width variations, and grain-size distribution. Finally, to provide practitioners with a
more quantitate criterion for step-pool channel design, measurements of the flow and sediment supply necessary to destabilize
steps should be taken and tested.

## 5 Conclusions

We reported results from flume experiments conducted to study the effect of sediment supply on the formation, evolution
and stability of steps in steep mountain streams subject to longitudinal width variations. Our feed experiments, together with
no-feed experiments previously conducted in the same flume (Saletti and Hassan, 2020), showed that more steps were created
when sediment was fed into the channel; however, the number of steps was inversely related to the feed rate suggesting that an
'optimum number of steps' is achieved for average values of sediment supply.



Steps formed in different locations due to different mechanisms and with different likelihoods. Steps in narrow and especially narrowing areas were more likely to form due to particle jamming and remain more stable than steps due to particle deposition around keystones in wide and widening areas. This was especially true at high water discharges, when steps in narrowing areas were the predominant morphological feature in all experiments. This has strong implications for stream restoration projects in steep-streams, where step-pool are often artificially designed to maintain channel stability while maintaining ecological functioning, especially during large floods.

The time-series of step formation, expansion/contraction and destruction showed that when sediment was fed into the channel, formation and destruction were the predominant means of channel evolution. However, when the supply was cut off, changes in existing steps (i.e., becoming wider or narrower) became very similar to step formation and destruction, indicating that the channel seemed to have achieved a more stable morphology. The difference between the trends of expansion/contraction versus formation/destruction was stronger for larger feed rates, conforming that sediment supply enhances particle activity and morphological changes.

The distributions of step survival showed that (a) steps in narrowing areas were more stable than those in other areas, and (b) channels subject to average sediment feed rates (i.e., 50% of the evaluated transport capacity) had steps that were more stable than those generated both without and with larger values of sediment feed. This last outcome, combined with step number results, led us to propose a conceptual model the relates step frequency to sediment supply in a functional way that resembles a bell curve. The maximum number of steps is achieved for average values of sediment supply; instead, the very low particle activity due to low sediment supply generates fewer steps, while the very high particle activity due to high sediment supply generates more steps that are more unstable.

Sediment yields measured at the channel outlet followed very closely the sediment inputs in all feed experiments, despite the very different values of feed rates spanning one order of magnitude. This brought us to question the applicability of the concept of transport capacity in steep channels, where the magnitude of sediment supply seem to be the first-order control on sediment transport rates.

Finally, combining data from feed and no-feed experiments, we tested the hypothesis proposed by Church and Zimmermann (2007) that a low sediment concentration is necessary to achieve step stability. Indeed, the fraction of unstable steps in our experiments clearly increased with sediment feed; this trend was clearer when evaluating the sediment concentration in terms of sediment yield (instead of sediment feed) which is a reasonable outcome, given that the sediment transport is a better proxy for what is happening in the channel.

Our results help to better understand the important role of sediment supply in the evolution of stepped channel morphology in steep streams, with the potential of being used by practitioners designing step-pool channels in stream restoration projects.

*Data availability.* The data used in this paper are available in the following public repository: https://doi.org/10.5281/zenodo.3754767.



*Author contributions.*  MS and MAH conceived the research and designed the experiment. MS carried out the experiments and analyzed the data under the supervision of MAH. Both authors wrote the paper.

*Competing interests.*  The authors declare no conflict of interest.

*Acknowledgements.*  MS was supported by a Swiss National Science Foundation (SNSF) Early.PostDoc Mobility grant (P2EZP2_172206)
and a Dean Scholarship awarded to MAH. MAH was supported by Natural Sciences and Engineering Research Council (NSERC) Discovery
and Canada Foundation for Innovation (CFI) grants. The authors thank Shawn M. Chartrand, J. Kevin Pierce and Conor McDowell for fruitful
discussions. Rick Ketler, David Waine, Conor McDowell and Yarra Hassan (who sieved most of the sediment) helped with the experiments.
J. Kevin Pierce and Conor McDowell provided careful reviews on a previous draft of this paper.




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
