# Peer review of "On how sediment supply affects step formation, evolution and stability in steep streams: an experimental study"

_Earth Surface Dynamics, 2020_

## Referee Comment (RC1) · Anonymous Referee #1 · 15 Jul 2020

Thank you for the opportunity to review this article. The subject matter is important and interesting, dealing with the question of how sediment supply into a steep stream channel influences step dynamics in terms of step frequency, location, and stability. Some important conclusions are drawn from the results, that can be of value to the scientific community and practical use for river restoration efforts. I suggest a few points to improve the clarity of the MS:

Line 88: instead of 'feed capacity' it would be better to use 'transport capacity' since this is the term used throughout the manuscript.

The paragraph starting from line 97 could be moved up (possibly before the paragraph

starting from line 84).

Line 99: Define $\psi$ here.

Table 1 and figure 3 both provide the same information, so I suggest using only the table which provides values of unit discharge and feed rates. If there is a practical need to include both the table and the figure, the legend in figure 3 should be boxed, as it looks like another sediment feed! Also in the figure caption, instead of 'In the last hours' it would be better to say 'after 7 hours'.

In table 1, if flow rates were increased by 20% every hour even after the sediment feed was stopped, why are the values same for some experiments (e.g. exp_07, exp_08, exp_09 in experiment 50, and exp_09 and exp_10 in experiment 100 and 150)?

In figures 3, 4, 5, 6, and 12, it is better to box the legend to avoid confusion.

Line 164: It would be helpful to the reader if you define jamming ratio here.

The paragraph starting from line 176: Correct the figure numbers referred in the 3 points.

Figure 7: Consider adding the results of the no-feed experiments also here, since it can provide a comparison between the step dynamics with and without sediment supply.

Line 258: Can you explain how your results can be used to elucidate the difference between channel stability and morphological stability in steep mountain streams?

The conclusion that 'the maximum number of steps is achieved for average values of sediment feed' can be misleading. If you consider the sediment feeds 50%, 100%, and 150% of the transport capacity, one can categorize them as low, average, and high sediment supplies, respectively. Then your conclusion implies that sediment feed corresponding to the 100% transport capacity (which is the average sediment feed in this categorization) creates the maximum number of steps, which is not true. Moreover, according to this categorization, it is actually the low sediment feed (not the average)

that creates the maximum number of steps. Therefore, I think the term 'average value' needs to be more explicitly stated. This is appropriately mentioned in line 287 where you have explained 'average value' in parenthesis (i.e., when the sediment feed is half of the transport capacity). But in other places including the abstract, the term 'average values of sediment supply' may lead to confusion.

Needs proofreading for grammatical errors and typos throughout the MS. Some examples are: Line 68: In hypothesis 3, qualitatively is mentioned twice Line 102: top of a moving chart ('of' is missing) Line 128: 'average number of' instead of 'average numbers of' The paragraph starting from line 155: check grammar in this paragraph. e.g. flow rates increases Figure 8 caption: '...that formed and destroyed' instead of 'an'. Line 226: '...as it is...' not '...at it is...' Line 340: '...conceptual model that relates...' not 'the relates'

---

## Referee Comment (RC2) · Anonymous Referee #2 · 22 Jul 2020

Overview

This paper presents the results of flume experiments investigating the effect of sediment supply on step-pool formation and stability. Despite I found it very similar to another paper recently publish by the authors (all results could have been published in a same paper?), the paper shows that there is an "optimum" level of sediment supply that maximizes the number of steps, which is a new result. The topic is interesting for field practice, and this work answers to a scientific question which has been previously formulated by several authors. The paper is very well written and all results are presented in a comprehensive manner, with appropriate figures. I suggest minor

revision.

Comments

I really regret that a paper focusing on step-pools does not presents one photo of this morphology observed in the flume. Like in Saletti and Hassan 2019, we must trust the author and what has detected the algorithm. It is really frustrating. Line 90: which did not you feed continuously? Line 85-95 and Fig11: there is no evidence that you were at capacity. It seems that all runs were under capacity? Figure 3: what are the 3 points at the bottom of the figure? Line 128: This results seems to contradict your hypothesis (Fig1)?

---

## Author Comment (AC1) · 24 Jul 2020

We thank both reviewers for their comments. We provide below our answers (in red) explaining how we plan to revise the manuscript accordingly.

REFEREE 1

Thank you for the opportunity to review this article. The subject matter is important and interesting, dealing with the question of how sediment supply into a steep stream channel influences step dynamics in terms of step frequency, location, and stability. Some important conclusions are drawn from the results, that can be of value to the scientific community and practical use for river restoration efforts. I suggest a few points to improve the clarity of the MS:

Line 88: instead of 'feed capacity' it would be better to use 'transport capacity' since this is the term used throughout the manuscript.

We agree that any confusion with this terminology should be avoided. As the reviewer correctly points out transport capacity is the term to be used here.

The paragraph starting from line 97 could be moved up (possibly before the paragraph starting from line 84).

We will do that.

Line 99: Define ψ here.

We will do that.

Table 1 and figure 3 both provide the same information, so I suggest using only the table which provides values of unit discharge and feed rates. If there is a practical need to include both the table and the figure, the legend in figure 3 should be boxed, as it looks like another sediment feed! Also in the figure caption, instead of 'In the last hours' it would be better to say 'after 7 hours'.

We feel like the figure provides a more effective visual summary, while the table reports the precise values. We believe they should both be provided in the manuscript. We will revise all the figures to make sure the legend does not create confusion.

In table 1, if flow rates were increased by 20% every hour even after the sediment feed was stopped, why are the values same for some experiments (e.g. exp_07, exp_08, exp_09 in experiment 50, and exp_09 and exp_10 in experiment 100 and 150)?

The numbers of the Q increments are different because we increased them after one hour only if there was a change in the bed. Otherwise we applied the same flow rate for another hour. We will make sure to clarify this point in the revised manuscript.

In figures 3, 4, 5, 6, and 12, it is better to box the legend to avoid confusion.

We will revise all the figures to make sure the legend does not create confusion.

Line 164: It would be helpful to the reader if you define jamming ratio here.

We will do that.

The paragraph starting from line 176: Correct the figure numbers referred in the 3 points.

Thanks to the reviewer for catching this. We will double-check all references to the figures to make sure they are correct.

Figure 7: Consider adding the results of the no-feed experiments also here, since it can provide a comparison between the step dynamics with and without sediment supply.

The reason why we did not include the no feed experiments in this figure is that the values of Q-Qs are different. In Fig 7-8 we focus on the difference between feed vs no feed in the same experiment as a function of time. Since in the no-feed experiments sediment feed is always zero and Q values are different (i.e. they started at lower values of Q), the comparison in this case is not straightforward.

Line 258: Can you explain how your results can be used to elucidate the difference between channel stability and morphological stability in steep mountain streams?

We agree this point should be further explained/discussed. We will make sure to do that.

The conclusion that 'the maximum number of steps is achieved for average values of sediment feed' can be misleading. If you consider the sediment feeds 50%, 100%, and 150% of the transport capacity, one can categorize them as low, average, and high sediment supplies, respectively. Then your conclusion implies that sediment feed corresponding to the 100% transport capacity (which is the average sediment feed in this categorization) creates the maximum number of steps, which is not true. Moreover, according to this categorization, it is actually the low sediment feed (not the average) that creates the maximum number of steps. Therefore, I think the term 'average value' needs to be more explicitly stated. This is appropriately mentioned in line 287 where you have explained 'average value' in parenthesis (i.e., when the sediment feed is half of the transport capacity). But in other places including the abstract, the term 'average values of sediment supply' may lead to confusion.

We will revise the terminology to make sure it does not create any confusion. Using "average", "low" and "high" can be indeed misleading so we will refer to the comparison between sediment feed and transport capacity.

Needs proofreading for grammatical errors and typos throughout the MS. Some exam- ples are: Line 68: In hypothesis 3, qualitatively is mentioned twice Line 102: top of a moving chart ('of' is missing) Line 128: 'average number of' instead of 'average num- bers of' The paragraph starting from line 155: check grammar in this paragraph. e.g. flow rates increases Figure 8 caption: '. . .that formed and destroyed' instead of 'an'. Line 226: '...as it is. . .' not '. . .at it is. . .' Line 340: '. . .conceptual model that relates. . .' not 'the relates'

We will carefully revise the manuscript to correct these errors.

REFEREE 2

Overview

This paper presents the results of flume experiments investigating the effect of sed- iment supply on step-pool formation and stability. Despite I found it very similar to another paper recently publish by the authors (all results could have been published in a same paper?), the paper shows that there is an "optimum" level of sediment supply that maximizes the number of steps, which is a new result. The topic is interesting for field practice, and this work answers to a scientific question which has been pre- viously formulated by several authors. The paper is very well written and all results are presented in a comprehensive manner, with appropriate figures. I suggest minor revision.

Comments

I really regret that a paper focusing on step-pools does not presents one photo of this morphology observed in the flume. Like in Saletti and Hassan 2019, we must trust the author and what has detected the algorithm. It is really frustrating.

We are sorry the reviewer feels we did not provide enough evidence for the identified steps. In the supplementary material (link in the 'Assets' webpage and at the end of the manuscript) we published all DEM maps and photos taken during the experiments. The algorithm we used has been tested and validated with different field and laboratory step-pool channels (see Zimmermann at al., 2008; Golly et al., 2019; Saletti and Hassan, 2020) and it has been applied in a systematic way to the data in all these experiments in order to reduce subjectivity related to bedform identification.

Line 90: which did not you feed continuously?

The feed was continuous over the first 40 mins.

Line 85-95 and Fig11: there is no evidence that you were at capacity. It seems that all runs were under capacity

We used 'capacity' in the sense of the transport capacity obtained with the Wilcock and Crowe (2003) model. One of the points we raised in the paper is precisely that these capacity formulations should be used carefully in steep channels. We will make sure to clarify this better in the revised manuscript.

Figure 3: what are the 3 points at the bottom of the figure?

They represent values of increased discharge and no sediment feed (see Table 1).

Line 128: This results seems to contradict your hypothesis (Fig1)?

We do not understand in which sense this would contradict our hypothesis. As we showed further on (Fig. 4 and 13a) the relation between step frequency and sediment feed follows the qualitative conceptual model depicted in Figure 1.

---

## Author Response (AR1)

Dear Editor,

We would like to thank the two Referees for their comments, which helped us to improve the paper. We are resubmitting a manuscript which has been changed as follows:
- We changed the title to make it more effective (and avoid possible confusion related to the word 'affects').
- We thoroughly reviewed the language, especially with respect to the sentence structure and the terminology. We made sure all variables used are properly defined and we avoided as much as possible confusing wording (guided by the comments of the reviewers). The final manuscript has been proofread by a native speaker.
- We improved all the figures based on the reviewers' comments.

We believe the manuscript is now ready for publication in Earth Surface Dynamics.
We also provide below a detailed reply to the Referees' comments and a copy of the manuscript where all changes have been highlighted.

Best regards,

Matteo Saletti and Marwan A. Hassan

REFEREE 1
Thank you for the opportunity to review this article. The subject matter is important and interesting, dealing with the question of how sediment supply into a steep stream channel influences step dynamics in terms of step frequency, location, and stability. Some important conclusions are drawn from the results, that can be of value to the scientific community and practical use for river restoration efforts. I suggest a few points to improve the clarity of the MS:

Line 88: instead of 'feed capacity' it would be better to use 'transport capacity' since this is the term used throughout the manuscript.

We agree that this terminology was confusing and we made sure to avoid that. As the reviewer correctly points out transport capacity is the term to be used here.

The paragraph starting from line 97 could be moved up (possibly before the paragraph starting from line 84).

We did that.

Line 99: Define ψ here.

We did that.

Table 1 and figure 3 both provide the same information, so I suggest using only the table which provides values of unit discharge and feed rates. If there is a practical need to include

both the table and the figure, the legend in figure 3 should be boxed, as it looks like another sediment feed! Also in the figure caption, instead of 'In the last hours' it would be better to say 'after 7 hours'.

We feel like the figure provides a more effective visual summary, while the table reports the precise values that are also useful. We believe they should both be provided in the manuscript. If the Editor thinks there is too much overlap, we could move the Table in the supplementary material.
We revised all the figures to make sure the legends do not create confusion.

In table 1, if flow rates were increased by 20% every hour even after the sediment feed was stopped, why are the values same for some experiments (e.g. exp_07, exp_08, exp_09 in experiment 50, and exp_09 and exp_10 in experiment 100 and 150)?

The numbers of the Q increments are different because we increased them after one hour only if there was a change in the bed. Otherwise we applied the same flow rate for another hour. We now clarified this (line 97).

In figures 3, 4, 5, 6, and 12, it is better to box the legend to avoid confusion.

The legend has been put in a box in all figures.

Line 164: It would be helpful to the reader if you define jamming ratio here.

Done.

The paragraph starting from line 176: Correct the figure numbers referred in the 3 points.

Thanks to the reviewer for catching this. We double-checked all references to the figures to make sure they are correct.

Figure 7: Consider adding the results of the no-feed experiments also here, since it can provide a comparison between the step dynamics with and without sediment supply.

The reason why we did not include the no feed experiments in this figure is that the values of Q-Qs are different. In Fig 7-8 we focus on the difference between feed vs no feed in the same experiment as a function of time. Since in the no-feed experiments sediment input is always zero and Q values are different (i.e., they started at lower values of Q), the comparison in this case is not straightforward.

Line 258: Can you explain how your results can be used to elucidate the difference between channel stability and morphological stability in steep mountain streams?

We agree with the reviewer that this point was misleading, so we decided to remove the sentence to avoid confusion and we address the stability issue in the discussion.

The conclusion that 'the maximum number of steps is achieved for average values of sediment feed' can be misleading. If you consider the sediment feeds 50%, 100%, and 150% of the transport capacity, one can categorize them as low, average, and high sediment

supplies, respectively. Then your conclusion implies that sediment feed corresponding to the 100% transport capacity (which is the average sediment feed in this categorization) creates the maximum number of steps, which is not true. Moreover, according to this categorization, it is actually the low sediment feed (not the average) that creates the maximum number of steps. Therefore, I think the term 'average value' needs to be more explicitly stated. This is appropriately mentioned in line 287 where you have explained 'average value' in parenthesis (i.e., when the sediment feed is half of the transport capacity). But in other places including the abstract, the term 'average values of sediment supply' may lead to confusion.

We do agree the use of average/low/high is misleading, so we avoided using that. We now always refer to the comparison between sediment feed and transport capacity.

Needs proofreading for grammatical errors and typos throughout the MS. Some exam- ples are: Line 68: In hypothesis 3, qualitatively is mentioned twice Line 102: top of a moving chart ('of' is missing) Line 128: 'average number of' instead of 'average num- bers of' The paragraph starting from line 155: check grammar in this paragraph. e.g. flow rates increases Figure 8 caption: '. . .that formed and destroyed' instead of 'an'. Line 226: '...as it is. . .' not '. . .at it is. . .' Line 340: '. . .conceptual model that relates. . .' not 'the relates'

The manuscript has been carefully revised and proofread by a native speaker.

REFEREE 2

Overview

This paper presents the results of flume experiments investigating the effect of sed- iment supply on step-pool formation and stability. Despite I found it very similar to another paper recently publish by the authors (all results could have been published in a same paper?), the paper shows that there is an "optimum" level of sediment supply that maximizes the number of steps, which is a new result. The topic is interesting for field practice, and this work answers to a scientific question which has been pre- viously formulated by several authors. The paper is very well written and all results are presented in a comprehensive manner, with appropriate figures. I suggest minor revision.

Comments

I really regret that a paper focusing on step-pools does not presents one photo of this morphology observed in the flume. Like in Saletti and Hassan 2019, we must trust the author and what has detected the algorithm. It is really frustrating.

We are sorry the reviewer feels we did not provide enough evidence for the identified steps. In the supplementary material (link in the 'Assets' webpage and at the end of the manuscript) we published all DEM maps and photos taken during the experiments. The algorithm we used has been tested and validated with different field and laboratory step-pool channels (see Zimmermann at al., 2008; Golly et al., 2019; Saletti and Hassan, 2020) and it has been applied in a systematic way to the data in all these experiments in order to reduce subjectivity related to bedform identification.

Line 90: which did not you feed continuously?

They feed was continuous over the first 40 mins (line 94).

Line 85-95 and Fig11: there is no evidence that you were at capacity. It seems that all runs were under capacity

We used 'capacity' in the sense of the transport capacity obtained with the Wilcock and Crowe (2003) model. One of the points we raised in the paper is precisely that these capacity formulations should be used with care in steep channels.

Figure 3: what are the 3 points at the bottom of the figure?

They represent values of increased discharge and no sediment feed (see Table 1).

Line 128: This results seems to contradict your hypothesis (Fig1)?

We do not understand in which sense this would contradict our hypothesis. As we showed further on (e.g., Fig. 4 and Fig. 13a) 
[revised manuscript text omitted]